# Effect of a Fourth Dose of mRNA Vaccine and of Immunosuppression in Preventing SARS-CoV-2 Breakthrough Infections in Heart Transplant Patients

**DOI:** 10.3390/microorganisms11030755

**Published:** 2023-03-15

**Authors:** Marco Masetti, Maria Francesca Scuppa, Alessio Aloisio, Laura Giovannini, Laura Borgese, Stefania Manno, Beatrice Tazza, Renato Pascale, Cecilia Bonazzetti, Natascia Caroccia, Mario Sabatino, Giosafat Spitaleri, Pierluigi Viale, Maddalena Giannella, Luciano Potena

**Affiliations:** 1Heart Failure and Transplant Unit, IRCCS Azienda Ospedaliero-Universitaria di Bologna, 40138 Bologna, Italy; marco.masetti5@unibo.it (M.M.);; 2Infectious Diseases Unit, IRCCS Azienda Ospedaliero-Universitaria di Bologna, 40138 Bologna, Italy; 3Department of Medical and Surgical Sciences, Alma Mater Studiorum-University of Bologna, 40138 Bologna, Italy

**Keywords:** COVID-19, heart transplantation, mRNA vaccines, fourth dose, immunosuppression, breakthrough infections

## Abstract

Patients with heart transplantation (HT) have an increased risk of COVID-19 disease and the efficacy of vaccines on antibody induction is lower, even after three or four doses. The aim of our study was to assess the efficacy of four doses on infections and their interplay with immunosuppression. We included in this retrospective study all adult HT patients (12/21–11/22) without prior infection receiving a third or fourth dose of mRNA vaccine. The endpoints were infections and the combined incidence of ICU hospitalizations/death after the last dose (6-month survival rate). Among 268 patients, 62 had an infection, and 27.3% received four doses. Following multivariate analysis, three vs. four doses, mycophenolate (MMF) therapy, and HT < 5 years were associated with an increased risk of infection. MMF ≥ 2000 mg/day independently predicted infection, together with the other variables, and was associated with ICU hospitalization/death. Patients on MMF had lower levels of anti-RBD antibodies, and a positive antibody response after the third dose was associated with a lower probability of infection. In HT patients, a fourth dose of vaccine against SARS-CoV-2 reduces the risk of infection at six months. Mycophenolate, particularly at high doses, reduces the clinical effectiveness of the fourth dose and the antibody response to the vaccine.

## 1. Introduction

Solid organ transplant (SOT) recipients are at higher risk of severe COVID-19 disease and related mortality [1], and they have therefore been prioritized worldwide for receiving mRNA-based vaccines against SARS-CoV-2 infection. Nevertheless, the immunosuppressed status of these patients has been reported to reduce the immune response to the vaccine, leaving transplant recipients at a higher risk of COVID-19 than the general population, despite a full course of vaccines [2,3,4]. The worldwide diffusion of the Omicron variant has highlighted the need for booster vaccine doses in order to achieve effective protection from severe COVID-19 in the general population, due to its increased resistance to the immunization induced by vaccines. It has been reported by large studies that a third and fourth dose compared with a second and a third dose of vaccine, respectively, can increase the induction of anti-receptor binding domain (RBD) IgG [5,6].

The assessment of anti-RBD antibodies can be used to assess the effectiveness of the immune response to a vaccine; however, this does not account for T-mediated immunity, which has been recently shown to have a role in the immunization against SARS-CoV-2.

Recent studies have shown that additional booster doses (third and fourth dose) can also increase levels of anti-RBD antibodies in heart transplant (HT) recipients [7,8], in the actuarial context of the Omicron variant, but at a lower degree when compared with the general population. We recently showed in the ORCHESTRA study that in SOT recipients, the levels of anti-RBD antibodies after vaccination are lower compared with healthy subjects [9].

Immunosuppression and, in particular, antiproliferative drugs have been suggested to be important mediators of a reduced response to vaccination in these patients, but available data are limited and are mainly derived from studies in liver or kidney transplantation; moreover, they are based on the evaluation of levels of anti-RBD antibodies rather than on clinical endpoints such as breakthrough infections or COVID-19-related hospitalization rates and death.

The aim of our study was to compare the efficacy of the third and fourth dose in preventing SARS-CoV-2 breakthrough infections and related hospitalizations and death during the current Omicron period, according to maintenance immunosuppression and to different patient-related risk factors.

## 2. Materials and Methods

### 2.1. Study Design

This report is a subset of the CONTRAST study, a single-center prospective study of SOT recipients who underwent SARS-CoV-2 vaccination. The study was approved by the local institutional review board (n° 167/2021/Oss/AOUBo on 12 March 2021). All included patients provided written informed consent. The study was conducted in accordance with the Declaration of Helsinki.

Herein we include all adult (>18 years) HT recipients seen for at least one clinical visit at our center between December 2021 and November 2022, for whom information about vaccination and SARS-CoV-2 infection was available. In this period, the Omicron variant was prevalent in Italy according to the data of the National Ministry of Health [10]. We included in this analysis all patients with either three or four doses of SARS-CoV-2 mRNA-based vaccine, and compared the occurrence of SARS-CoV-2 infection between these two patient groups.

Patients experiencing SARS-CoV-2 infection between the third and fourth dose, for the purposes of this analysis, were included in the three-doses group. Because the immune protection from mRNA vaccines has been reported to be effective after at least two weeks following vaccine administration [11,12], patient follow-up started two weeks after the last vaccine dose. This approach allowed us to avoid overlaps between the two study groups.

We excluded those who had COVID-19 prior to the third dose and those whose vaccination cycle was started or completed before heart transplantation, because these conditions may represent confounding factors in evaluating the clinical efficacy of vaccine doses in the context of immunosuppressed patients [13]. We also excluded those with incomplete information about vaccination doses, uncertainties about COVID-19 episodes, and congenital heart disease before HT.

As per clinical practice, all patients were encouraged to receive mRNA vaccination at least three months after HT. Vaccines became available on a broad scale in Italy in January 2021 and the third (booster) dose in September of the same year, while the fourth dose became available in February 2022.

We collected data about demography, main comorbidities, time from transplantation, immunosuppression at the administration of the last vaccine dose (immunosuppressive drug type and dose, reported as mg/day, and through levels of tacrolimus, cyclosporine or everolimus, reported as ng/mL), and the number of vaccine doses during the Omicron period.

The standard immunosuppression regimen in our center consists of induction therapy with thymoglobulines after surgery and maintenance long-term therapy, based on an antiproliferative agent (mycophenolate as a first-choice agent, and mammalian target of rapamycin (mTOR) inhibitors as the second line), a calcineurin inhibitor (tacrolimus or cyclosporine) and steroids. The steroid dose is progressively reduced early after HT if no rejection is detected in the routine endomyocardial biopsies. In the last five years, tacrolimus has replaced cyclosporine as a first-line calcineurin inhibitor in our center.

SARS-CoV-2 infection was defined by the positivity of antigenic and/or molecular assays in nasopharyngeal swabs. Patients were encouraged to test for SARS-CoV-2 in cases of COVID-19-compatible symptoms or after a known contact with a positive subject. In addition, a subset of patients undergoing screening surveillance endomyocardial biopsies were tested before accessing the catheterization laboratory, as per policy in our hospital. All included patients were interviewed about the previous occurrence of COVID-19 infection or symptoms suggestive of infection during the scheduled visits within the study period and at the end of the follow-up (30 November 2022).

### 2.2. SARS-CoV-2 Antibody Testing

Antibody positivity was determined by Elecsys^®^ Anti-SARS-CoV2 ECLIA assay (Roche Diagnostics AG, Rotkreuz, Switzerland) and defined according to the receptor-binding domain (RBD) IgG titer. Minimum and maximum thresholds for the detection of anti-RBD antibody levels were 0.4 and 2500 UI/mL, respectively. A positive antibody response was defined as an anti-RBD titer ≥5 U/mL, as previously specified [9].

Patients were encouraged to undergo antibody testing after the third dose (after at least two months). The third dose was recommended to all patients, regardless of the levels of antibodies before its administration, when available, following the indications of the National Ministry of Health.

### 2.3. Study Endpoints

Study outcome measures were evaluated during the six-month period after the last vaccine dose. As the primary endpoint we analyzed the survival free from breakthrough SARS-CoV-2 infection, defined by the detection of a positive molecular or antigenic assay for SARS-CoV-2 in nasopharyngeal swab; as the secondary endpoint we considered the survival free from the combined incidence of hospitalization in the intensive care unit (ICU) or death from COVID-19. Both of these outcomes are expressed as survival rates at six months after the last dose of vaccine received.

We also evaluated the relationship between anti-RBD antibodies after the third dose, immunosuppression and subsequent infection occurrence in the subset of patients for which these data were available.

### 2.4. Statistical Analysis

Continuous variables were expressed as the mean ± standard deviation if normally distributed, or as the median and interquartile range (IQR) if non-normally distributed, and categorical variables were expressed as a number (percentage). Differences between groups were tested according to t-Student, Wilcoxon or Pearson’s test, for continuous or categorical variables, as appropriate. A *p* ≤ 0.05 was considered as significant. Survival free from the study outcome measures was estimated by the Kaplan–Meier method. For a small subset of patients (*n* = 17) receiving tixagevimab/cilgavimab as prophylaxis after the third or fourth dose, censoring took place at the moment of its administration. Cox’s proportional hazards test was performed to identify risk factors for the study endpoints. Variables significantly associated with study outcomes following univariate analysis were included in a multivariate model to identify independent predictors of breakthrough infections.

Statistical analysis was performed with SAS JMP 9.0 Software (SAS Institute Inc., Cary, NC, USA).

Considering the low number of deaths and/or admissions to the ICU, we did not perform a multivariate analysis for this endpoint; we investigated only the role of the variables that were shown to be independent predictors of infections.

## 3. Results

### 3.1. Study Population

In total, 285 patients met the inclusion criteria; 17 (5.9%) of them had COVID-19 before the third dose. Therefore, 268 patients constituted the overall study cohort; 15 (5.6%) of them were combined transplants (eight heart–liver transplantation, six heart–kidney, and one heart–liver–kidney).

The characteristics of the overall study population are depicted in Table 1; mean age at the time of the most recent vaccination was 61 years and the great majority of patients were male. Regarding immunosuppression, most patients (63.8%) were on therapy with mycophenolate (MMF) and more than 60% with steroids; cyclosporine was the most widely used calcineurin inhibitor (64.9%); and 60 patients (22%) had less than 5 years of post-transplant follow-up.

### 3.2. Survival Free from SARS-CoV-2 Events

In total, 62 patients developed a SARS-CoV-2 breakthrough infection within six months after the last dose of vaccine, accounting for a survival free from infection of 75.7 ± 2.7%; no one experienced a reinfection.

Seven patients were admitted to the ICU (11.3% of the infected), four of whom died (6.4% of the infected), accounting for an overall survival free from hospitalization in the ICU or death of 96.9 ± 1.1% at six months. A total of 28 patients did not receive any treatment for SARS-CoV-2 infection, 16 were prescribed antiviral drugs (6 molnupiravir, 6 remdesivir, and 4 nirmatrelvir), and 11 were prescribed sotrovimab; in 7 cases data were missing.

### 3.3. Vaccine Campaign and Its Effect on Infections and Mortality

Among 268 patients, 195 (72.7%) received the third dose and 73 (27.3%) the fourth. The third dose was given 6.0 ± 3.3 months after the second dose and at a mean time of 12.3 ± 7.4 years after HT; the fourth dose was given 5.9 ± 1.1 months after the third. Patients receiving the fourth dose were generally older (Table 2), without any other statistically significant differences compared with those vaccinated with three doses. Among 62 infections, 53 (85.5%) occurred in patients vaccinated with three doses, and 9 (14.5%) in those with four doses; patients receiving the fourth dose had a significantly higher survival free from breakthrough infections at 6 months compared with those who had received only the third dose (85.9 ± 4.4% vs. 72.5 ± 3.2%, four vs. three, *p* = 0.03, Figure 1A).

The secondary endpoint occurred in seven patients; five of them had received three doses. All patients who died had received only three doses. Overall, the survival free from COVID-19-related ICU hospitalization and death was similar between four and three doses (96.7 ± 2.3% vs. 97.0 ± 1.3%, four vs. three, *p* = 0.86, Figure 1B).

### 3.4. Factors Influencing Clinical Response to Vaccines

Table 3 shows the different characteristics of patients who became infected compared with those who were not infected. Breakthrough infections occurred more frequently in patients who were on therapy with mycophenolate at the time of the third dose (82.7 ± 3.9% vs. 72.0 ± 3.5%, no MMF vs. MMF, *p* = 0.05, Figure 2A); in particular, the survival free from infection was significantly lower in patients taking a dose of MMF above the upper quartile (≥2000 mg/day) of the overall population (82.7 ± 3.9% vs. 74.5 ± 4.4% vs. 68.2 ± 5.7% no MMF vs. <2000 mg/day vs. ≥2000 mg/day, *p* = 0.05, Figure 2B). Survival free from infection was also lower in patients who had received HT in the previous 5 years (65.6 ± 6.2% vs. 78.8 ± 2.9%, ≤5 years vs. >5 years, *p* = 0.02, Figure 2C).

We did not find any other predictor of infection following univariate analysis; in particular, older age was not associated with an increased risk of infection, nor was the use other immunosuppressive drugs or comorbidities.

According to the results of multivariate analysis (Table 4A), vaccination with only three doses, therapy with MMF, and a time from HT shorter than five years were independently associated with a higher risk of being infected by SARS-CoV-2, even after adjusting for age (*p* < 0.05 for all). When factoring the dose of MMF into the multivariable model (Table 4B), we found an intake ≥2000 mg/day to be an independent predictor of a higher chance of being infected, together with the other variables already specified above in the text.

Patients on therapy with MMF or taking a dose ≥2000 mg/day also experienced a lower survival rate from SARS-CoV-2-related deaths or admission to the ICU (100% vs. 95.0 ± 1.8%, no MMF vs. MMF, *p* = 0.04; 100% vs.97.7 ± 1.6% vs. 90.5 ± 4.1%, no MMF vs. ≤2000 mg/day vs. ≥2000 mg/day, *p* < 0.01, Figure 3A,B).

### 3.5. Anti-RBD Antibodies and Their Relationship with MMF and Breakthrough Infections

Data about anti-RBD antibodies were available for 183 of 195 patients who received only the third dose and were assessed at a mean time of 5 ± 2 months after vaccination; at the moment of the current analysis, we have only few data about antibodies after the fourth dose and therefore they have not been analyzed. The antibody values were not normally distributed; 47 patients (25.6%) had no anti-RBD antibody response according to the pre-specified threshold. In total, 18 patients had a SARS-CoV-2 infection between the third dose administration and antibody assessment; except for one case, all of them (94.4%) had a positive antibody response. Among the remaining 165 patients, we found that those experiencing an infection later in the follow-up (*n* = 31) had lower levels of antibodies after the third dose (15.9 (IQR 0.4-1160.0) vs. 764.4, (IQR 8.3-2500.0), infected vs. not infected, *p* = 0.04, Wilcoxon’s test) and less frequently a positive antibody response (52.2% vs. 75.4%, infected vs. not infected, *p* = 0.03). Patients without an antibody response had a higher risk of developing SARS-CoV-2 infection (survival free from infection at 6 months: 73.4 ± 6.9 vs. 89.7 ± 2.8%, negative vs. positive, *p* = 0.006, Figure 4). Linear regression analysis showed a correlation between the level of antibodies and the probability of infection, with a value of 185 U/mL identified by ROC analysis as the best threshold to predict the outcome (sensitivity: 65%, specificity: 63%, AUC: 0.64, *p* = 0.04). Unfortunately, we did not have any data about antibodies in patients who died or were admitted to the ICU; therefore, we are not able to analyze their effect on this endpoint.

Patients on therapy with MMF at the time of third dose administration had a significantly lower value of anti-RBD antibodies (385 (IQR 0.4-2500) vs. 1327, (IQR 79-2500), no MMF vs. MMF, *p* = 0.02) and less frequently a positive antibody response (65.8% vs. 84.6%, no MMF vs. MMF, *p* < 0.01). We did not find any association between other immunosuppressive drugs or other comorbidities and levels of antibodies after the third dose.

## 4. Discussion

In this study, we compared the effect of the fourth and the third dose of mRNA vaccine against SARS-CoV-2 on breakthrough infection rate, COVID-19-related hospitalizations in the ICU and death at six months in heart transplant recipients without a previous COVID-19 infection in the current context of the Omicron variant.

Our main findings are as follows: (1) while the fourth dose led to a higher protection against breakthrough infections, its efficacy appeared to be strongly influenced by MMF, particularly when taken at high doses (≥2000 mg/day); (2) patients transplanted for less than 5 years have a higher risk of infection; (3) MMF intake was associated with an increased risk of death and ICU hospitalization; and (4) levels of anti-RBD antibodies measured after the third dose were influenced by MMF intake and predicted the subsequent risk of infection.

It is well known from the general population that two doses of vaccine reduce the risk of SARS-CoV-2 infection and strongly decrease the related risk of hospitalization and death [14]; however, the efficacy of this approach is lower in SOT recipients than in the general population, and, among them, HT recipients have been reported to have the lowest anti-RBD antibody titers after vaccination [15,16]. The Omicron variant is especially resistant to vaccines and can elude vaccine-induced immunity. In this context, a third and a fourth dose of vaccine can increase anti-RBD antibodies and the cellular immune response to SARS-CoV-2 compared with a placebo and reduce the risk of breakthrough infections, COVID-19-related hospitalizations, and death, both in the general population [5,6] and in HT patients [16,17,18]. Therefore, additional doses of vaccine have been recommended to HT recipients [19]. However, a consistent number of patients, including those with HT, can still experience breakthrough infections even after the fourth dose.

Our study confirms, in a larger sample, previous data about the clinical efficacy of the fourth dose (Peled and co-workers reported two series of only 90 and 96 patients [7,8]). In addition, in our work, we not only report anti-RBD or neutralizing antibodies as outcome measures [20], but also the clinical endpoint of the occurrence of breakthrough infections and related serious adverse events, which we believe is more adherent to clinical practice, where the main objective is to avoid infections and hospitalizations.

Patients with SOT who become infected with SARS-CoV-2 can experience a higher rate of serious complications compared with the general population; this could lead clinicians to hospitalize them even when they are in less sick conditions. To avoid this potential bias, we restricted our outcomes to ICU hospitalization and death. We found that Omicron variants are resistant to repeated doses of vaccine in HT recipients, because, even after four doses, a consistent amount of them show a mortality rate higher than in the general population, although considerably lower than that reported before vaccines became available, when it was close to 30% [21].

These findings support the need to identify factors that can potentially influence the response to vaccines.

Immunosuppression is the key player in influencing the immune response to vaccines in SOT recipients, and our results provide insights into its effect, highlighting the role of MMF.

Our most important novel finding is that mycophenolate can reduce the clinical effectiveness of the fourth dose on the occurrence of infections, and that its role is particularly pronounced at doses higher than or equal to 2000 mg/day. Moreover, it is noteworthy that this effect was observed also when we analyzed ICU hospitalizations and death, and that all patients who were hospitalized or who died were taking ≥2000 mg/day of MMF, even if the number of events was low.

Other Authors have investigated the role of MMF in response to vaccines. Mitchell [22] and Ben Zadok [23] showed in heart and lung transplant recipients that MMF > 1000 mg/day can be associated with reduced anti-RBD antibodies after two doses of vaccine; Peled and colleagues [7] found similar results both after the second and third dose in a larger cohort. In a large study involving more than 800 kidney transplant recipients [24], MMF was associated with lower anti-RBD and neutralizing antibodies after three doses of vaccine. A recent metanalysis [25] including 83 studies in SOT recipients found older age and therapy with antimetabolites to be predictors of a poor humoral response to three doses of vaccines. A study including recipients of thoracic transplants [26], of whom 134 were heart-transplanted patients, showed that the levels of antibodies after a second dose were decreased in patients receiving antiproliferative agents such as MMF.

By analyzing a substantially larger sample size, we found that MMF can impair not only the antibody response but also the clinical efficacy of vaccines. We also demonstrated an inverse relationship between antibody response after the third dose and the subsequent occurrence of SARS-CoV-2 infection, corroborating the weak findings available so far [27], especially for the Omicron variant, which seems to be less sensitive to anti-RBD antibody neutralization [28].

MMF is an antiproliferative drug blocking mainly T (and partly B) cells’ proliferation [29]. Our findings therefore suggest its involvement in the T-mediated immunity induced by the vaccines, not only in antibody production. Recent studies [30] highlighted the importance of both T-mediated immunity and the humoral response; Peled et al. [7] and a recent work by Hall et al. [31] showed that some patients without an antibody response after vaccine can still have a positive T-mediated response, mainly based on CD4+ T cells. Unfortunately, in our study, data exploring the role of SARS-CoV-2-specific T-mediated immunity were unavailable.

Taken altogether, our results suggest that the reduction in MMF dose, especially in patients with a poor immune response after two doses, recently transplanted and at low risk of rejection, may help in achieving an immune response induced by the vaccine. These findings could have relevant clinical implications; according to the ISHLT Registry [32], about 85% of transplanted patients are on MMF one year after transplant. In our cohort, its use was lower (65.3%) because a consistent number of long-term patients were on mTOR inhibitors as a rescue therapy for the prevention of cardiac allograft vasculopathy. A small non-randomized small monocentric study in liver and kidney transplantation showed that in patients who were not on MMF at the moment of second dose or in those in whom the drug was withdrawn before its administration, the titer of antibodies was higher and there was an inverse relationship with the serum concentration of MMF [33].

Patients who had received a transplant in the previous five years had an increased risk of infection, similar to the findings of Hallett and contributors [26]. Although difficult to prove, it can be speculated that this could be related to the higher global immunosuppressive burden of the early period after HT. Even if we were not able to show an effect of other immunosuppressive drugs on breakthrough infections, the observed higher infection rates in patients transplanted in a more recent era do not allow us to completely exclude a potential role of other immunosuppressive therapies.

Our study has some limitations. First, we did not genotype the virus, and therefore had to extrapolate that infections were related to Omicron variants based on the epidemiological prevalence in the general population during the study period. Second, we may not have captured all infected or hospitalized cases of COVID-19 for several reasons (60% of patients managed by our center lived far from our region; others were asymptomatic); however, this is a limitation of most clinical trials of SARS-CoV-2, and patients were interviewed about SARS-CoV-2 infection episodes at each clinical visit. Third, we did not analyze the data of antibodies after the fourth dose, because this was available for only a few patients at the moment of the current analysis, and we do not have any data about antibody levels in patients who were hospitalized or died as a result of COVID-19. Fourth, it was not possible to measure adherence to medical therapy, nor to obtain information about patients’ behaviors that could have potentially influenced the results (i.e., poor compliance with mask wearing in public places). Finally, we were not able to perform a functional test of immune activity (i.e., Elispot), and did not have complete available data on lymphocyte count or other potential conditions that put patients at high risk of infection, such as hypogammaglobulinemia.

Another aspect to be considered is that patients who did not experience a breakthrough infection up to the fourth dose might have an intrinsic lower risk of infection than those who were infected after the third dose, irrespective of the vaccine dose per se. However, this bias is difficult to be eliminated in a study comparing the effectiveness of add-on cycles of therapies, because time is an intrinsic risk factor that cannot be adjusted for; our approach was coherent with other population-based studies [34]. The clinical characteristics between the three- and four-dose group were not different, except for age, which was higher in the four-dose group and, although not significant in our study, is a known risk factor for SARS-CoV-2 infection in the general population. In light of these considerations, the different risk of infection between the two groups was unlikely to be ascribed to a selection bias, and the observed results can be largely attributed to the efficacy of the fourth dose.

The observed number of ICU hospitalizations and deaths could have been influenced by the variations in the prevalence of infection in the general population according to the different timepoints of follow-up; however, we were not able to analyze this relationship because of the low number of events and the existence of some overlap between the administration of the third and fourth dose in our cohort in the same time period.

As previously specified, seventeen patients received an additional prophylaxis with tixagevimab/cilgavimab: fourteen after the third dose, and three after the fourth. To avoid any potential confounding effect, the follow-up was censored at the moment of T/C administration.

Despite these limitations, this is the first study, to the best of our knowledge, comparing the interplay between the fourth dose of mRNA vaccine, immunosuppression, and other variables in a large population of heart transplant recipients using clinical endpoints related to SARS-CoV-2 infection.

Future studies are needed to evaluate the efficacy of a potential reduction in MMF prior to vaccine doses on positive antibody response, T-mediated immunity, and rates of infection and hospitalization.

## 5. Conclusions

In the current context of Omicron as a prevalent variant, in heart transplant recipients without previous SARS-CoV-2 infection, the fourth dose of mRNA vaccine gives higher protection against breakthrough infections at six months as compared to the third dose. Therapy with mycophenolate, especially when given at high doses (≥2000 mg/day), influences the clinical efficacy of vaccine doses, increasing the risk of breakthrough infections, hospitalizations, and death, and reducing the occurrence of a positive antibody response after the third dose. Patients that recently underwent a transplant are at particularly high risk. Levels of anti-RBD antibodies after vaccination seem to be strongly correlated with the subsequent risk of infection. Our results support the efforts to find approaches to the prevention of SARS-CoV-2 infection that are different from the strategy of only repeated booster doses to protect this fragile category of patients, and suggest the evaluation of a reduction in mycophenolate before the administration of a booster dose. Further studies are needed to confirm the efficacy and safety of this latter assumption.

## Figures and Tables

**Figure 1 microorganisms-11-00755-f001:**
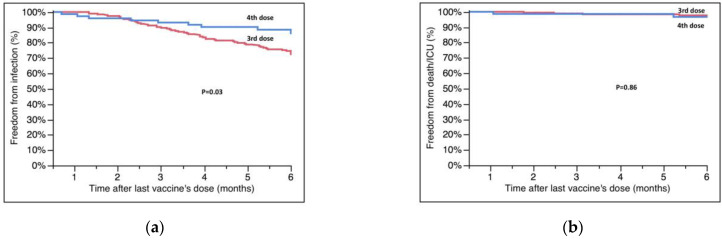
(**a**) Survival free from SARS-CoV-2 infection at six months according to the number of vaccine doses. (**b**) Survival free from ICU hospitalization/death according to the number of vaccine doses. Survival rates are reported in the text.

**Figure 2 microorganisms-11-00755-f002:**
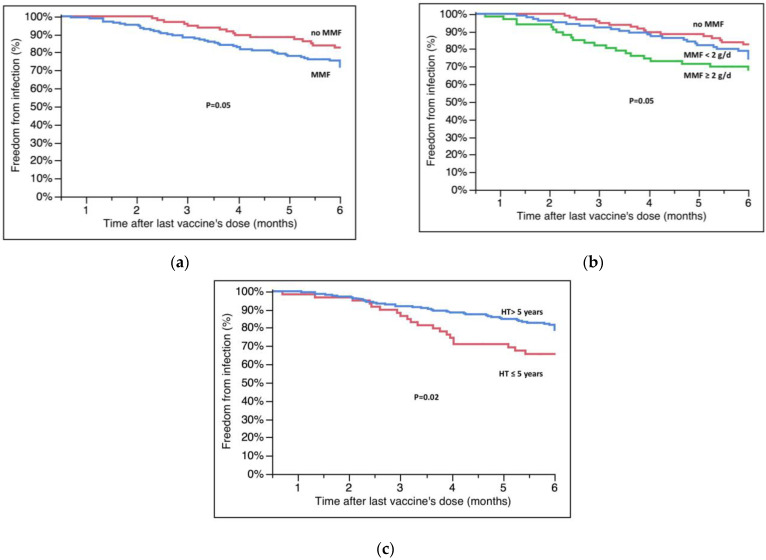
(**a**) Survival free from infection according to MMF therapy. (**b**) Survival free from infection according to dose of MMF at the moment of last vaccination. (**c**) Effect of time from HT on infections. Survival rates are reported in the text; g/d = grams/day.

**Figure 3 microorganisms-11-00755-f003:**
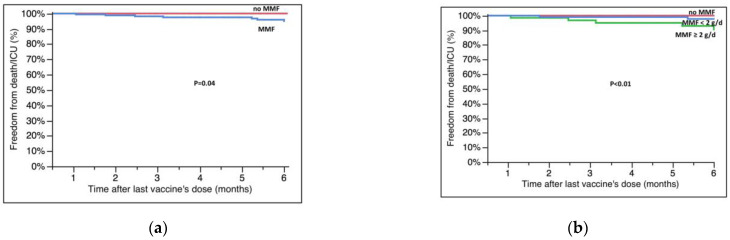
Survival free from hospitalization in the ICU/death at six months after the last dose of vaccine according to (**a**) MMF therapy and (**b**) dose of MMF (at the moment of vaccination in both cases). Survival rates are reported in the text; g/d = grams/day.

**Figure 4 microorganisms-11-00755-f004:**
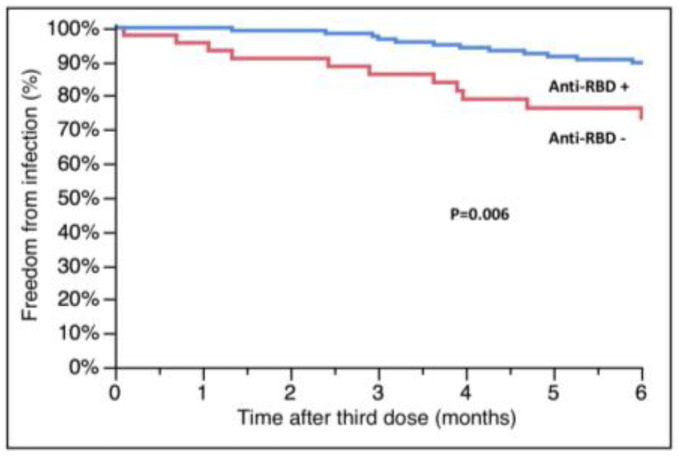
Relationship between anti-RBD antibodies after the third dose of vaccine and subsequent SARS-CoV-2 infection. RBD: receptor-binding domain.

**Table 1 microorganisms-11-00755-t001:** Characteristics of the study population.

Variable ^1^	(*n* = 268)
**Demographic**	
Age at last vaccine dose (years)	61.4 ± 12.8
Gender (males), *n* (%)	198 (73.8%)
Distance from HT (years)	12.3 ± 7.3
Distance from HT ≤ 5 years, *n* (%)	60 (22.4%)
**Immunosuppression, *n* (%)**	
Antiproliferative agents	234 (87.3%)
MMF	171 (63.8%)
Everolimus	57 (21.3%)
Azatioprine	6 (2.2%)
Calcineurin inhibitors	267 (99.6%)
Cyclosporine	174 (64.9%)
Tacrolimus	93 (34.7%)
Steroids	169 (63.0%)
Steroid dose > 5 mg/day	39 (14.6%)
**Comorbidities, *n* (%)**	
Hypertension	184 (69.1%)
Diabetes	83 (30.9%)
Renal failure (GFR < 60 mL/min)	166 (61.9%)
BMI > 30, *n* (%)	32 (11.9%)

^1^ at the moment of the last vaccine dose.

**Table 2 microorganisms-11-00755-t002:** Characteristics of the individuals receiving three vs. four doses of vaccine.

Variable	Three Doses (*n* = 195)	Four Doses (*n* = 73)	*p*
**Demographic**			
Age at last vaccine dose (years)	60.3 ± 12.9	64.9 ± 11.6	0.008
Gender (males)	142 (72.8%)	56 (76.7%)	0.51
Distance from HT (years)	12.3 ± 7.3	12.4 ± 7.1	0.85
Distance from HT ≤ 5 years, *n* (%)	43 (22.1%)	17 (23.3%)	0.82
**Immunosuppression**			
MMF, *n* (%)	126 (64.6%)	45 (61.6%)	0.22
MMF dose			0.63
No MMF	69 (35.4%)	28 (38.4%)	
MMF < 2000 mg/day	79 (40.5%)	25 (34.2%)	
MMF ≥ 2000 mg/day	47 (24.1%)	20 (27.4%)	
Everolimus, *n* (%)	44 (22.8%)	13 (17.8%)	
Calcineurin inhibitors, *n* (%)	195 (100%)	72 (99.6%)	0.22
Cyclosporine	125 (64.4%)	49 (67.1%)	
Tacrolimus	70 (35.6%)	23 (31.5%)	
Steroids, *n* (%)	129 (66.8%)	40 (54.8%)	0.07
Steroids dose > 5 mg/day	31 (16.0%)	8 (11.0%)	0.30
**Comorbidities, *n* (%)**			
Hypertension	129 (66.8%)	55 (76.4%)	0.12
Diabetes	61 (31.3%)	22 (30.6%)	0.90
Renal failure (GFR < 60 mL/min)	118 (60.5%)	48 (65.7%)	0.71
BMI > 30	23 (11.8%)	9 (12.5%)	0.87

**Table 3 microorganisms-11-00755-t003:** Different characteristics of the study population according to known SARS-CoV-2 infection.

Variable	Infected (*n* = 62)	Non Infected (*n* = 206)	*p*
**Vaccine doses**			0.007
Three doses, *n* (%)	53 (85.5%)	141 (68.8%)	
Four doses, *n* (%)	9 (14.5%)	64 (31.2%)	
**Demographic**			
Age at last vaccine dose (years)	60.5 ± 13.4	61.6 ± 12.7	0.57
Gender (males)	45 (72.6%)	153 (74.3%)	0.79
Distance from HT (years)	10.7 ± 7.4	12.7 ± 7.3	0.05
Distance from HT ≤ 5 years	20 (32.3%)	40 (19.4%)	0.03
**Immunosuppression**			
MMF, *n* (%)	46 (74.2%)	125 (60.7%)	0.04
MMF dose			0.05
No MMF	16 (25.8%)	81 (39.3%)	
MMF < 2000 mg/day	25 (40.3%)	79 (38.3%)	
MMF ≥ 2000 mg/day	21 (33.8%)	46 (22.3%)	
Everolimus, *n* (%)	11 (17.7%)	46 (22.3%)	0.45
Calcineurin inhibitors, *n* (%)	62 (100%)	205 (99.5%)	0.79
Cyclosporine	42 (67.7%)	133 (64.5%)	
Tacrolimus	20 (32.3%)	72 (34.9%)	
Steroids, n (%)	37 (59.7%)	132 (64.1%)	0.59
Steroids dose > 5 mg/day	12 (19.4%)	27 (13.1%)	0.23
**Comorbidities, *n* (%)**			
Hypertension	44 (71.0%)	140 (68.3%)	0.72
Diabetes	23 (37.1%)	60 (29.3%)	0.25
Renal failure (GFR < 60 mL/min)	40 (67.8%)	128 (69.2%)	0.71
BMI > 30	7 (11.3%)	25 (12.2%)	0.85

**Table 4 microorganisms-11-00755-t004:** (**A**) Multivariate analysis for the endpoint of SARS-CoV-2 infection 6 months after the last dose of vaccine (model including MMF but not its dose). (**B**) Multivariate analysis for the endpoint of SARS-CoV-2 infection 6 months after the last dose of vaccine (model including MMF dose).

**A**		
**Variable**	**HR (95% CI)**	** *p* **
Vaccine doses (four vs. three)	0.47 (0.21–0.91)	0.02
Distance from HT ≤ 5 years	1.97 (1.12–3.37)	0.02
Mycophenolate	1.76 (1.02–3.20)	0.04
Age at last vaccine dose	1.24 (0.98–1.02)	0.99
**B**		
**Variable**	**HR (95% CI)**	** *p* **
Vaccine doses (four vs. three)	0.45 (0.20–0.89)	0.02
Distance from HT ≤ 5 years	1.93 (1.09–3.30)	0.02
Mycophenolate dose ≥ 2000 mg/day	2.22 (1.16–4.32)	0.02
Age at last vaccine dose	1.01 (0.98–1.02)	0.98

HR: hazard ratio; CI: confidence intervals.

## Data Availability

The data presented in this study are available on request from the corresponding Author. The data are not publicly available due to GDPR regulations because they are part of the CONTRAST clinical study.

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
