# Peer review of "Effect of a Fourth Dose of mRNA Vaccine and of Immunosuppression in Preventing SARS-CoV-2 Breakthrough Infections in Heart Transplant Patients"

_microorganisms, 2023, doi:10.3390/microorganisms11030755_

Round 1

Reviewer 1 Report

The authors assessed the efficacy of a fourth dose of Covid-19 mRNA vaccination in heart transplant recipients. Strengths are large number of heart transplant recipients included and the follow-up for clinical endpoints. The main limitation is the confirmatory nature of the data.

I have some concerns and comments that the authors may want to address.

-          The risk for Covid-19 can be also reduced by the use of preventive anti-spike monoclonal antibodies (Evusheld). Can you describe how many patients (if any) have received Evusheld? Also for the risk factors for severe Covid-19, therapeutic strategies for early infection such as the use of monoclonal antibodies (sotrovimab) and antivirals (remdesivir, nirmatrelvir) needs to be described.

-          I would better specify the design of the study: when the patients were included, when samples for immunogenicity were obtained (baseline, after vaccination), how patients were followed-up for assessing of infection, etc. It is not entirely clear in the current form.

-          One methodological issue that I see with the current presentation of the results is that the two compared groups (3 vs. 4 doses) are not completely independent, as they are actually the same cohort of patients taken in two different time-points. Patients in the “3-dose group” at higher risk for covid might be actually excluded from the “4-dose group” because they have actually developed covid (so that they are censored and additionally did not have the opportunity to receive the 4th dose in the time frame of the study). By definition, these patients are a lower risk for Covid. Please comment on that.

-          Data on immunogenicity is tricky: there should be very few patients that developed Covid after the determination of the antibody levels up to six months post vaccination. How the authors have determined the predictive values of the antibody levels?

-          Of the 17 patients excluded for previous Covid, were all of them positive by PCR/antigens or also had a positive anti-NC serology?

-          Sometimes it is confusing to use the cumulative incidence and the free-event survival for the same variables.  

-          Discussion can be more focused.

Minor issues

-          The usual term is “maintenance” not “background” immunosuppression

-          The sentence “positivity of antigenic and/or molecular swabs” is not accurate. It should be “positivity of antigenic and/or molecular assays in nasopharyngeal swabs” (and other respiratory samples, I guess).

-          “According to this, actuarial recommendations suggest”, should be “current recommendations”

Reviewer 2 Report

Interesting article regarding the effect of revaccination against SARS-Cov2 in heart recipients. I wonder if the authors have available data about the presence of lymphopenia or IgG hypogammaglobulinemia at the time of evaluation to assess the potential role of these factors as covariates.

Reviewer 3 Report

Interesting study about the comparison between 3 and 4 doses of clinical efficacy on SOT. you have suggested some interesting findings, I have a few questions/suggestions

- need spell check and there are inappropriate capitalizations of letters that need corrections: eg Hypertension is spelled wrong. 

- Methods: Are subjects who received the 4th dose a subset of those who received the 3rd dose? it is not clear from the methods or if they are separate.

- Also, The study period is over 1 year, have you looked into if the external factors like wave or R0 of disease had any effect on hospitalization/death?

-Discussion: I felt was too long. It could be concise, especially the MMF part.  Also, thymoglobulin part, is there a need to mention it since it's too far away from the dose/ surgery? 

Round 2

Reviewer 1 Report

The authors have appropriately addessed my concerns. As a minor comment, maybe the discussion continues to be a little too long. 

Author Response

Thank you for your positive comments and appreciation of our work. We are happy that we have addressed your concerns. According to your suggestion, we further significantly shortened the discussion of more than 400 words (see the attached tracked file). Thank you.

Reviewer 2 Report

The authors have answered to this reviewer.

Author Response

Thank you for your positive comments.